# Multi-Omics Mining of lncRNAs with Biological and Clinical Relevance in Cancer

**DOI:** 10.3390/ijms242316600

**Published:** 2023-11-22

**Authors:** Ivan Salido-Guadarrama, Sandra L. Romero-Cordoba, Bertha Rueda-Zarazua

**Affiliations:** 1Departamento de Bioinformatìca y Análisis Estadísticos, Instituto Nacional de Perinatología Isidro Espinosa de los Reyes, Mexico City 11000, Mexico; 2Departamento de Medicina Genómica y Toxicología Ambiental, Instituto de Investigaciones Biomédicas, Universidad Nacional Autónoma de México, Mexico City 04510, Mexico; sromero@iibiomedicas.unam.mx; 3Biochemistry Department, Instituto Nacional de Ciencias Médicas y Nutrición Salvador Zubirán, Mexico City 14080, Mexico; 4Posgrado en Ciencias Biológicas, Facultad de Medicina, Universidad Nacional Autónoma de México, Mexico City 04510, Mexico; bgruedazara@gmail.com

**Keywords:** long non-coding RNAs, cancer, multi-omics

## Abstract

In this review, we provide a general overview of the current panorama of mining strategies for multi-omics data to investigate lncRNAs with an actual or potential role as biological markers in cancer. Several multi-omics studies focusing on lncRNAs have been performed in the past with varying scopes. Nevertheless, many questions remain regarding the pragmatic application of different molecular technologies and bioinformatics algorithms for mining multi-omics data. Here, we attempt to address some of the less discussed aspects of the practical applications using different study designs for incorporating bioinformatics and statistical analyses of multi-omics data. Finally, we discuss the potential improvements and new paradigms aimed at unraveling the role and utility of lncRNAs in cancer and their potential use as molecular markers for cancer diagnosis and outcome prediction.

## 1. Introduction

The advent of next-generation sequencing and other revolutionary technologies for the study of omics, such as genomics, transcriptomics, proteomics, and metabolomics, has completely transformed the way basic and clinical cancer research is conducted [1,2]. As a consequence of these fast-moving advances, we have been able to produce large volumes of data from an assortment of multiple omics layers (i.e., genomics, epigenomics, transcriptomics, proteomics, metabolomics, and microbiomics) from different tissue and cell models for the study of basic and translational research in cancer [3,4]. In particular, the exponential advances in sequencing technology have allowed researchers to explore the complexity and diversity of human transcriptomes [5,6]. One population of RNA molecules that has received special attention in recent years are long non-coding RNAs (lncRNAs), which are molecules without protein-coding capacities but with versatile and pleiotropic functions [7,8], with well-defined action mechanisms linked to different hallmarks of cancer [9,10,11]. lncRNAs are increasingly being recognized as potential pharmacological targets and as diagnostic and prognostic biomarkers [12,13,14]. Indeed, advances in RNA-seq and other large-scale methodologies represent a valuable resource for deepening our knowledge of the molecular aspects of lncRNAs and their roles in cancer biology [15,16]. So far, most studies have primarily centered on detecting aberrant expression changes of lncRNAs. However, given the complex and pleiotropic functions of lncRNAs, we must recognize that to truly harness the potential of these transcripts to reveal their complex functional aspects and associations with extrinsic and intrinsic factors that influence the susceptibility, incidence, and survival of different types of cancer, we need to incorporate new frameworks for multidimensional analyses, as have been proposed for the study of other complex diseases [17]. In this context, multi-omics analysis approaches that combine and integrate protein-coding gene expressions with other dimensions of molecular information, such as genomics and epigenomic, have improved our capacity for obtaining significant understanding of the molecular aberrations underpinning the oncologic characteristics in different types and subtypes of tumors (e.g., cancer cell fate and survival) [4,18]. Likewise, a multi-omics framework progressively incorporated into research is helping us to delineate the way lncRNAs perform their functions and how they impact cancer development [19]. As a new paradigm, these approaches might boost research to produce sufficient evidence to ascribe them a causal relationship to the malignancy presence and most importantly, a clinical value as diagnostics, prognostic, and predictive markers in cancer [20,21].

In this review we do not intend to cover an exhaustive compendium of the many lncRNAs and their biological and molecular roles in cancer, nor of the large number of bioinformatics programs that have been developed for the analysis of transcriptomics, genomics, and epigenomics data, both of which have been extensively discussed elsewhere [18,22,23,24]. Instead, we will try to shed light on some of the less discussed aspects of the practical applications using different integrative, comparative, and multidimensional study designs incorporating bioinformatics and statistical analysis of multi-omics data. Finally, we will discuss the possible improvements and new paradigms aimed at unraveling the activity of lncRNAs in cancer, and their potential use as molecular markers in clinical and biological behaviors. Throughout the following sections, we present and discuss the recent scientific evidence employing different omics strategies for the functional and biological study of lncRNAs in cancer research, with special interest in their informative role in oncology programs. Next, we focus our discussion on machine learning and biostatistics solutions that have been incorporated as part of methodological pipelines for quantifying the performance of lncRNAs signatures in the classification of cancer samples and the prediction of response and survival in cancer patients. Finally, we discuss future scenarios of new technological advances for the discovery and profiling of lncRNAs in cancer, and the challenges of translating these advances into real-world clinical settings.

The complexity of genomic alterations, gene expression, protein–protein interactions, epigenetic mechanisms, cellular secretome, and metabolome is a complex subject, especially when we want to define how their components interact (Figure 1). For this reason, and considering our main interest, in this work, we searched for literature in the PubMed database using the keywords (“lncRNA” or “long non-coding RNA” and “cancer” and “multi-omics”) up to October 2023. Subsequently, we manually revised our search to focus mainly on the different approaches for multi-omics integration with the scope of finding and evaluating functional lncRNAs signatures with a clinical prognosis and prediction of patient outcome. The integration of the data can occur from a simple to a more complex view of cellular function. We will start with the simplest view, which focuses on describing the combination of phenotypes observed at different biological levels independently.

## 2. lncRNAs Associated with Cancer Driver Somatic Mutations Profiles

Cancer is characterized by the somatic acquisition of various cellular alterations that lead to selective advantages, such as unrestricted growth, the suppression of apoptosis, and enhanced metabolism [25], and genomic instability has been widely recognized as one of the leading hallmarks of cancer [26]. Recently, especially using high-throughput sequencing, research has revealed the complexity of the somatic DNA aberration landscapes of cancer genomes [27]. In fact, defining specific somatic mutational profiles has long been a keystone for characterizing cancer patients and leading to the discovery of specific cancer driver genes [28], which in turn can help estimate the inherited risk, prioritize therapy and prognosis, and understand pharmacologic responses to drug treatments [29]. Not surprisingly, given the protein-centric view that has prevailed in many branches of molecular biology in recent years, cancer studies have focused more on exploring the mutations that occur in protein-coding regions, despite the fact that the exome represents a very small fraction (1–2%) of the human genome (Figure 1). Excitingly, with the expansion of our knowledge about other functional regions within the human genome other than the regions with protein-coding loci, a growing number of discoveries have now emerged demonstrating that recurrent somatic mutations in non-protein-coding regions can also be informative features, paving the way for discovering causal links between lncRNAs functions and oncogenesis [30]. By leveraging whole-genome sequencing data, researchers have been able to better characterize the presence, frequency, and functional impacts of somatic mutations in the non-coding regions of cancer genomes [31]. To our current knowledge, the most robust evidence has shown that cancer driver point mutations and structural variants are less common in non-coding genes and regulatory sequences than in protein-coding genes. The availability of whole-genome sequencing data and capture panels that include non-coding regulatory regions from projects such as TCGA, along with other valuable efforts, has led to the discovery of driving somatic mutations in non-coding regions that affect the function of genes other than the protein-coding genes [32,33,34,35]. For example, Rheinbay et al. sequenced 360 primary breast tumors and patient-matched normal samples, and observed cancer driver mutations in the regulatory non-coding regions that could indeed occur at frequencies similar to the coding regions. However, they also found promoter elements with a significant burden or clustering of mutations, including promoters of lncRNA genes, such as RMRP8 and NEAT1. Notably, three of the four mutations present in the NEAT1 promoter element induced a reduction in the NEAT1 expression, thus adding new loss-of-function alterations affecting NEAT1 in breast cancer [36]. This study represents one of the most sophisticated integrative efforts performed to date. The approach used by the authors emphasized the discovery of somatic alterations that did not occur at the loci encoding the transcripts of lncRNAs, but in the regulatory regions that controlled their transcription (Figure 2). A broader view of the state-of-the-art approaches that integrate multiple omics data to characterize the somatic point mutations at loci-encoding lncRNAs that may lead to a gain or a loss of function are discussed in Table 1.

## 3. Approaches for Prioritizing Cancer Driver lncRNAs Using Somatic Mutations Profiles

From an omics and bioinformatics point of view, we can broadly classify the approaches that have been developed to prioritize lncRNAs with a possible association with cancer into at least two categories: (1) an early integrative analysis of somatic variants with cancer driver potential that have an impact on the functional activity of lncRNAs associated with prognosis in cancer patients [37,38,69,70,71], and (2) post-analysis integration strategies centered on the relationship between genome instability and lncRNAs with aberrant expressions associated with tumor prognosis [39,40,41,42,43,44,45] (Figure 2). A representative study was conducted to evaluate the potential impact of somatic mutations in human lncRNAs (denominated MutLncs) and their functional significance in cancer, interrogate the mutation profiles in genomic regions harboring lncRNA and their vicinity across 17 cancer types, and use an integrative pipeline to describe the significance of the MutLncs contribution to cancer [69]. Interestingly, they discovered different cancer-specific networks of co-occurring lncRNAs in the MutLncs by considering the mixing effects (i.e., the influence of a pairwise combination of multidimensional data, including DNA methylation, TF expression, and miRNA expression). Notably, the study found that a number of MutLncs in co-occurring pairs were correlated with patient survival, particularly in human cutaneous squamous cell carcinoma (SKCM) and glioblastoma (GBM) tumors [69].

The complexity of cancer genomes goes beyond a set of specific targetable driver mutations. Other features, such as the tumor mutational burden (TMB) and mutational signatures, are increasingly being incorporated into bioinformatic analyses aimed at uncovering therapeutic and prognostic insights in cancer [72]. Therefore, it is not unexpected that many studies have integrated somatic mutation profiles and transcriptome expression data to identify lncRNAs [39,40,41,42,46] (Table 1). For example, in one study, colon cancer researchers used a post-analysis integration strategy to characterize patients by their somatic mutation profiles and subsequently analyzed the transcriptome data and survival information from the same set of patients. They were able to derive a score calculated on a seven genome instability-associated lncRNA signature that was able to distinguish high-risk patients characterized by high somatic mutation, high microsatellite instability, significantly worse clinical outcomes, and specific tumor immune infiltration [43]. Using a similar approach in cervical endometrial cancer (CESC), Zhang and collaborators created a computational model derived from what they called the mutator hypothesis, in which they first grouped patients based on the cumulative number of somatic mutations, then screened them for differences in the lncRNAs expression, and found a genomic instability-associated lncRNAs signature with the potential to determine patient survival and help differentiate high-risk from low-risk CESC tumors [44]. Remarkably, the TMB has become an emerging marker for checkpoint inhibitor-based immunotherapy [44,73]. In a recent study, by integrating RNA sequencing data, mutation profiles, and the corresponding clinical information from a TCGA colorectal cancer dataset (COAD), the authors evaluated the TMB of the COAD samples to classify them into low- and high-TMB groups. They, then, used a machine learning approach to construct a 14-lncRNA-based classifier index related to three immune checkpoints (i.e., PD1, PD-L1, and CTLA-4). Interestingly, the obtained classifier was significantly associated with the TMB levels and could accurately predict the overall survival of the COAD patients [74].

The number of studies discussed so far provided a vision of different modalities of integrative multi-omics analyses being employed as powerful strategies for identifying and characterizing lncRNAs as informative features that allow for diagnosis, prediction, and prognosis. Without an integrative genomic approach, it is unlikely that more complex questions can be addressed, for example, the search for the interaction of lncRNAs with other biological and environmental factors, such as sex hormones, smoking, occupational risk, and dietary habits, among others, and their association with the risk of cancer development, patient survival, and response to treatment. For instance, lung adenocarcinoma (LADC), the most common type of lung cancer, is largely caused by chronic tobacco smoking. However, approximately 25% of cases occur in non-smokers. Notably, this proportion of LADC in non-smokers is associated with being female, and the etiology remains an elusive question. In a recent study to uncover genomic evidence of smoking-associated mutagenesis, researchers performed a multilevel integrative analysis using whole-genome sequencing and RNA-seq data to unravel the mutational processes and complex genomic rearrangements that drive the development of LADC. One of the intriguing findings of this multi-omics analysis was the discovery of a recurrent fusion gene, composed of the gene ERBB3 and the lncRNA anti-estrogen-resistance 4 (BCAR4), which was present in two tumors from female cases in which low to no contribution of a mutational signature associated with direct DNA damage caused by tobacco smoke mutagens (COSMIC signature 4) was observed [75] (Figure 3). Recently, other comprehensive integrative genomic analyses have led to the discovery of diverse fusion variants involving BCAR4 in lung adenocarcinoma [76,77,78]. Interestingly, the CD63–BCAR4 fusion was discovered in a never-smoking female patient by another genome-wide study on non-small cell lung cancer (NSCLC) [79]. Here, the authors found that the expression of both transcripts within this fusion were highly activated. Considering that a higher expression of BCAR4 was also established as an independent predictive factor for tamoxifen resistance and poor progression-free survival in ER+ breast cancer patients [80], the clinical relevance of these complex rearrangements involving BCAR4 in other tumor types, such as lung cancer, may become more apparent and demonstrate the utility of using highly multidimensional molecular analyses to uncover the targetable alterations that would otherwise be overlooked.

## 4. Copy-Number Alterations in Genomic Regions Encoding lncRNAs

Indeed, genomics and mathematical analyses of the patterns of somatic alterations have become a powerful strategy for identifying cancer driver genes. In this regard, although somatic point mutations and small insertions and deletions (INDELs) have been the primary focus of cancer genomic studies, copy-number alterations (CNAs) are also important forms of DNA aberrations, also referred to as somatic copy-number alterations (SCNAs), which encompass larger genomic regions and often harbor key genes involved in the development and progression of many cancers [81,82] (Figure 1). Not surprisingly, most large-scale genomic analyses conducted to date have successfully identified almost exclusively protein-coding cancer driver genes located in regions of focal amplification and deletion [83,84,85], and it is only relatively recently that the first systematic analyses have reported on the identification of lncRNAs that are also contained within focal CNAs in cancer genomes [47,86,87,88,89]. Given the evidence that nearly three-quarters of the human genome can be transcribed to RNA [90] and that only 2% of the human transcriptional landscape codes for a protein, the need to understand the functional impact of SCNAs on lncRNAs has become clearer. In one of the first comprehensive characterizations of the impact of cancer driver lncRNAs in regions of SCNA alteration, Hu et al. implemented a bioinformatic integration of SCNAs and an expression analysis across 12 different cancer types. In developing their approach, the researchers scanned large-scale genomic data for focal alterations, which often exhibited high-amplitude variations, and mapped the known lncRNA loci to these regions of a focal gain or loss. Interestingly, approx. 17.8% of the lncRNAs-encoded loci with SCNAs were expressed in 40 cancer cell lines representing five different tumor types, among which they discovered FAL1 as a potentially oncogenic lncRNA associated with clinical outcomes in ovarian cancer patients. In a follow-up study, the authors confirmed the SCNAs in FAL1 in OC and observed that it was also present in the other five tumor types [48,49,50,82]. The authors also raised a key issue to consider in the characterization of lncRNAs as tumorigenic drivers by distinguishing the driver SCNAs from passenger ones by their levels of presence (i.e., in sufficient quantities to be detected) in the cancer cells of interest. In this regard, the identification of SCNAs from the lncRNA loci that are actively transcribed in a given cell at a given time may help characterize the lncRNA molecules with a real function and role as drivers of tumorigenesis and of potential clinical utility, as further demonstrated by Hu et al. and other studies [47,48,51,54,55,56,87] (Table 1).

A second consideration that arose in light of the above approaches for understanding the actual contribution of lncRNAs as drivers of cancer and its deregulation was based on estimating the relationship that exists between the copy-number values of the SCNAs at the lncRNA loci and the expression levels of their corresponding transcripts present in a given cancer type. Indeed, this aspect was consistent with our knowledge that the differential gene expressions of the protein-coding cancer driver genes were significantly correlated with the CNAs [82,91]. As a consequence, researchers have begun to delineate the correlation between the SCNAs and expression levels as useful parameters for defining the gain or loss of function of lncRNAs with oncogenic or tumor suppressor functions and for potentially associating them with the clinical outcome for different tumors [47,48,49,50,56,57,82]. 

Cancer is a heterogeneous disease that includes a diversity of tumors from the same or different organs and tissues, displaying differences in cellular compositions and biological and molecular features [92]. Much of our understanding about cancer heterogeneity and its clinical implications has been improved by new developments in genomics [93]. Some studies have so far addressed the integration of multi-omics data, including the lncRNAs expression and SCNA, with the aim of defining the molecular profiles that capture the diversity of the tumor subtypes with different prognoses and responses to treatment [58,59,94]. As a notable example, in a large-scale genomic analysis of high-grade serous ovarian carcinoma (OV), Akrami et al. investigated the correlation between SCNA and lncRNAs expression profiles and were able to identify the lncRNA OVAL (RP11-522D2.1) as a target of focal DNA amplification. Notably, this alteration was detected specifically in OV and was consistent with the increase in the OVAL transcript expression [95]. It is important to note that this relationship may be more complex, and that the SCNAs present in a tumor may be associated with aberrant expressions of lncRNAs other than those encoded at their loci. For example, other mechanisms may operate to disrupt their activity, such as the occurrence of a breakpoint event within the boundaries of the promoters and regulatory sites of a particular lncRNA, or genomic alterations that directly disrupt the protein-coding gene targets for such lncRNAs. In this regard, further research is still needed to unravel the complicated interactions between lncRNAs, protein-coding genes, miRNAs, and other types of functional elements that are potential drivers of cancer.

## 5. The Revolution of Non-Coding Transcriptome in Cancer Studies

There is increasing evidence that large numbers of lncRNAs are present in the human genome. The latest release of LNCipedia, a public database for lncRNA sequences and annotation, contains 107,039 high-confidence transcripts that show no coding potential belonging to 49,372 lncRNA genes [96]. Undoubtedly, the burgeoning number of lncRNAs over the past few years has aroused the interest of the scientific community in the study of their biological properties and roles [11]. Based on the fact that they play critical roles in various processes that are necessary for body cell functions, many studies have addressed and demonstrated the relationship between lncRNAs and disease [21]. In particular, experimental evidence for the relevance of lncRNAs as cancer markers has been accumulating over the last decades. Among the most prominent examples of lncRNAs whose aberrant expression is associated with cancer development are the descriptions of MALAT1 [97], HOTAIR [98], and PCA3 [99]. Since its discovery, MALAT1 has become the paradigm of functional alterations of lncRNA in cancer and has been proposed as a potential biomarker with a critical role in several tumors, such as non-small cell lung adenocarcinoma (NSCLC) [100], gastric cancer [101], and colorectal cancer [102]. The oncogenic expression of lncRNA HOTAIR also holds a promising value as a biomarker of the response to breast [103] and hepatocellular carcinoma [104]. Meanwhile, the levels of PCA3 in urine has been used as a marker for prostate cancer aggressiveness to guide medical decisions on patient treatment and has been available since its approval by the FDA [105,106]. Aside from PCA3, no other lncRNA has been approved by the FDA for diagnostic, prognostic, or treatment response prediction in any type of cancer. Despite these successful examples discovering more associations between lncRNAs and diseases, characterizing them as biomarkers or targets of therapeutic agents has become an increasingly challenging task. It is at this point that the integrative analysis of large-scale data has become a game changer. In this regard, there are several scientific examples that have used analytical strategies to investigate the relationship between somatic alterations, including somatic mutations, copy-number alteration profiles, and transcriptome alterations, such as the perturbed correlated expression of protein-coding genes, microRNAs, and lncRNAs, to reveal cancer drivers or biomarkers with potential utility in the prognosis and prediction of the treatment response. As shown in Table 1, we enumerated many of the most relevant examples involving multi-omics strategies for studying the biology and clinical utility of lncRNAs. In the following sections, we will review some of them in more detail to provide insights into these studies and exemplify the cross-linking, integration, and complementation between the different traits that were analyzed using multiple omics dimensions.

## 6. Multi-Omics Network Approaches Reveal lncRNA Biological Relevance on Cancer Biology

As discussed above, the technological and computational advances in recent decades have allowed for the identification of thousands of lncRNAs whose molecular alterations are associated with various types of cancer (Figure 1). A major challenge is that the initial efforts to discover cancer-related lncRNAs took advantage of classical functional genomic approaches, primarily by characterizing the global transcriptomic landscape, evolutionary conservation, or proximity to known cancer genes. While these analytical strategies provided valuable insights, the altered transcriptional profiles alone do not indicate a causal role in cancer programs. Currently, various biological evidence describing known cancer-associated lncRNAs has been presented in the literature, most of which focused on the function of a single lncRNA linked to malignant transformation through its role in gene regulation and its impact on cancer hallmarks.

Despite intense research into the mechanisms underlying altered lncRNAs, our understanding of the global biological impact of lncRNA regulation in tumors remains limited [107]. It is challenging to holistically describe the complex biological processes of many lncRNAs and their cooperative mechanisms using traditional biochemical and molecular approaches. Therefore, the availability of multi-omics data from genomics (mutations and CNV), transcriptomics (mRNA, non-coding RNA), proteomics, epigenetics (chromatin methylation and architecture), and metabolomic studies has opened new strategies for advancing the understanding of the roles of lncRNAs in health and disease by developing tools to integrate these data at the systems level (Figure 1).

In this section, we will review the progress made in understanding the multi-omics biological features through machine learning and artificial intelligence approaches and provide an overview of the recent advances in uncovering the regulatory basis underlying the functionalities of lncRNAs at various molecular and cellular biological levels.

### Describing the Novel lncRNAs Drivers in Cancer through Multi-Omics Integration

Apart from a few well-known lncRNAs, the landscape of cancer lncRNAs is far from complete. The diverse functional repertoire of lncRNAs in cancer can be explored through (1) their function as driver genes, resulting from early mutations that are positively selected during tumorigenesis, or (2) as downstream genes, resulting from non-genetic changes in their expression, localization, or molecular interactions [108]. Although both categories contribute to cancer phenotypes, most efforts to discover cancer lncRNAs only take advantage of differential expression approaches. To overcome this, several computational methods have recently been developed to identify lncRNA driver genes by analyzing coordinate omics alterations to detect the signals of positive selection.

One of the first statistical methods for driver gene discovery was OncodriveFML [109], which identifies tumor-associated lncRNAs by interrogating somatic mutations in the coding and non-coding regions and gene expression. Compared to the other methods, OncodriveFML calculates a functional impact score, for which it uses a local mutational background in specific regions to define the positive selection signals in genes across tumor tissues. The method takes into account that the mutational background is influenced by chromatin architecture, replication timing, and transcription factor binding sites. Therefore, by considering this local background as well as the mutational and expression patterns, this computational tool can discover lncRNAs contained in potential genomic driver regions involved in tumorigenesis. Computing this method using whole-genome tumor data sequenced by the TCGA and the Cancer Genome Project, it was possible to identify MALAT and MIAT as two lncRNAs with an excess of high-impact mutations.

Another integrative method for predicting lncRNA drivers that are relevant to tumorigenesis is ExInAtor, which identifies genes with a high somatic single nucleotide variant load across the tumor genomes using the DNA mutation patterns (local trinucleotide background model) and expression data as proxies for their functionality. ExInAtor was modeled on 1112 whole genomes from 23 cancer types deposited in GENCODE, and predicted 15 lncRNA drivers with a high confidence, of which nine were novel lncRNAs and six were known cancer-related transcripts, including PCA3, MALAT1, BCAR4, lncRNA-ATB, and SAMMSON. Most of these lncRNAs were tumor-specific, although NEAT1 and MALAT1 were identified in a pan-cancer context, confirming their role in tumorigenesis. The set of previously unreported driver lncRNAs include MIR100HG, AP000469.2, RP11-308N19.1, RP11-455B3.1, RP11-332J15.1, RP11-707A18.1, RP11-6c14.1, RP11-1101K5.1, RP11-354A14.1, and RP11-189E14.4. These novel candidates are evolutionarily conserved, expressed in normal tissues, and have an increased gene length. They also tend to be proximal to cancer SNPs and are encoded in CNA regions, suggesting a role in tumorigenesis. The authors highlighted MIR100HG, which was highly conserved and had canonical histone modifications in the promoter region and transcription factor binding sites [110].

More recently, to enhance the discovery of cancer-related lncRNAs and gain insight into their biology, the Cancer lncRNA Census (CLC) was presented as a tool to provide functional or genetic evidence of lncRNAs roles in cancer by integrating genomic and transcriptomic data associated with cancer in different mammalian species [108]. To date, it is not completely clear whether mutated lncRNAs can drive tumorigenesis and whether such altered functions could be conserved during evolution. Therefore, the CLC considers the conserved functions between humans and mice as a relevant feature that could provide strong evidence for the biological role of lncRNAs, both in cancer and under physiological conditions. The application of this computational model revealed the colocalization of cancer lncRNAs with known protein-coding cancer genes. A total of 10 tumor-causing mutations were identified in eight lncRNA orthologs, including DLEU2, GAS5, MONC, NEAT1, PINT, PVT1, SLNCR1, and XIS, some of which have already been reported in cancer.

The integration of DNA, RNA, and protein alterations and the way they cooperatively interact provide new evidence for identifying dysregulated lncRNA in cancer. For example, LongHorn [111], a recently presented computational method, integrates genomic, transcriptomic, and proteomic alterations and predicts dysregulated lncRNA regulatory networks in cancer pathways by modeling their impact on the transcription factors, RNA binding proteins and microRNAs activity, lncRNA promoter binding sites, and post-transcriptional activation/inhibition. The computation of this method for 14 cancer types from the TCGA predicted several lncRNA candidates whose dysregulation affected other known cancer genes and pathways mainly in a tumor-specific context and influenced tumor etiology. OIP5-AS1, TUG1, NEAT1, MALAT1, XIST, and TSIX were predicted to regulate cancer signaling in multiple tumor contexts. In addition, the lncRNA network analyses indicated the enrichment of lncRNA binding sites in the promoter regions of messenger RNAs, which enhanced the transcriptional effects of lncRNAs. The functional experimental analyses confirmed most of the predictions made with LongHorn.

An interesting approach presented by Mitra and collaborators for predicting the biological dependencies of uncharacterized lncRNAs focused on the identification of co-essential modules by integrating the copy number, and epigenetic and transcriptomic data of the lncRNA landscape from exogenous knockouts or activation screens that were generated using CRISPR techniques [112]. By applying this model to multi-omics cancer cell line data, the authors estimated 289 lncRNA-gene co-expression networks that recapitulated the known proliferation-regulating lncRNAs and predicted novel lncRNAs associated with proliferative signaling that were poorly characterized, such as PSLR-1/2, which induced G2 arrest through the modulation of the FOXM1 transcriptional network and whose exogenous expression inhibited proliferation and colony formation in cell line models.

Although DNA methylation dysregulation is associated with cancer, the molecular mechanisms of how methylation and transcriptional lncRNA patterns are reciprocally modulated in cancer remain largely unknown. A novel integrative analysis framework, called MeLncTRN (Methylation mediated lncRNA Transcriptional Regulatory Network), integrates transcriptome, DNA methylome, and copy-number variation profiles to identify the regulatory circuits driven by epigenetically driven lncRNAs across 18 cancer types [113]. An analysis of 5970 TCGA tumor samples revealed that the association between lncRNAs and the DNA methylation mechanisms was common and conserved across multiple cancer types, e.g., a complex interplay between lncRNAs and epigenetic modulators, such as DNA cytosine methyltransferases DNMT1, and histone modification proteins, such as EZH2 (Figure 4). For example, FAM83H-AS1, TUG1, PVT1, and LINC00511 acted as scaffolds to enhance EZH2 or DNMT1 binding and consequently repressed the expression of their mRNA targets. This observation expands the understanding of the role of lncRNAs in transcriptional regulatory circuits in addition to their miRNA sponge activity as competitive endogenous RNA (ceRNA) [114].

Emerging evidence has also revealed the underlying crosstalk between lncRNA and genomic instability, a relevant hallmark of cancer. Novel approaches integrating chip-seq, WGS, and WES data revealed an unexpected relationship between oncogenic lncRNAs and epigenetic alterations that contribute to chromosome fragility in cancer. To characterize the lncRNA-based mechanism by which aberrant epigenetic signatures can be generated, the authors used as a conceptual, the sub-telomeric chromosomal locus 8q24, which contains the cMYC gene and a large histone domain H3 (CENP-A) variant, both of which were altered in cancer cells of various solid tumors. This region also encoded five unique lncRNAs sequences, namely PCAT1, PCAT2, CCAT1, CCAT2, and PVT1, which negatively modulated the occupancy of CENP-A at the chromosomal locus. Their results indicated a competition between the lncRNAs transcription and R-loop occupancy that strongly contributed to the maintenance of CENP-A invasion, which ultimately affected the chromosome stability [115].

An oncology milestone that was already mentioned is the TMB, which is related to the infiltration of various immune cell populations that can enhance or limit cancer programs [116]. Recently, increasing evidence has shown that lncRNAs may play fundamental roles in the regulation of the immune system, but few immune-related lncRNAs have been described in cancer. Therefore, novel approaches have been developed to shed light on these roles [117,118]. Through an integrative analysis of the lncRNA expression, tumor immune response signatures, and genome-wide DNA methylation data in 9626 tumor samples across 32 cancer types, the lincRNA-based immune response (LIMER) tool revealed 7528 lincRNAs associated with the tumor immune signature [119]. Of interest, EPIC1 was identified as a relevant immune-related lncRNA that was inversely correlated with the MHC expression and CD8+ T activation and infiltration. The in vitro and in vivo models demonstrated that EPIC induced tumor immune evasion and resistance to immunotherapy by epigenetically suppressing the tumor cell antigen presentation through EZH2 interaction. Another interesting tool is ImmLnc, which systematically infers the candidate lncRNA modulators of the immune-related pathways by matching the gene and lncRNA expression profiles. The tool prioritizes cancer-related lncRNAs by comprehensively characterizing the lncRNA landscape and its correlation with the immunome. One of the first findings was that the tumors derived from similar tissues were likely to share lncRNA immune regulators. Furthermore, the novel subtypes identified by ImmLnc showed a distinct mutation burden, immune cell infiltration, expression of immunomodulatory genes, and response to chemotherapy and prognosis [117].

## 7. Challenges and Opportunities of Machine Learning to Deepen the Functional and Biological Roles of lncRNAs in Cancer

In this section, we provide a summary of the approaches we took to address biological data integration and discuss the pitfalls of overcoming and optimizing the use of data from multiple sources and technologies in an analysis pipeline. A first challenge was the difference between the samples, replicates, and technology used, which favored batch effects. Additionally, the different modalities for measuring the cell levels with different measurements posed a significant computational challenge, as there was no common virtual space to integrate the samples.

Machine learning methods have been increasingly used in attempts to better understand the biology of molecules, such as lncRNAs. These methods can be generally classified into supervised and unsupervised learning. The former includes dimension reduction and clustering, while the latter includes regression and classification. In the context of the lncRNA landscape, one of the challenges that has arisen is the accurate identification of true lncRNAs from other types of RNAs, along with the definition of their biological roles. The main approach for the identification of lncRNAs is based on their coding potential and length. Later, other features were integrated, such as the presence of an open reading frame (ORF), nucleotide composition, kmers, secondary structure, codon usage, ribosome release score, conservation scores, and others. Among the most used algorithms for data integration are logistic regression, random forest (RF), support vector machine (SVM), and deep learning (Figure 2). From a mechanistic point of view, to better understand the role of lncRNAs in complex diseases such as cancer, in addition to the coding potential, some studies included computational frameworks based on machine learning methods, such as the Genetic Importance Calculator (GIC), which considers the interaction of lncRNAs with other omics levels. Through these methods, it is possible to accurately identify unknown lncRNA interactions. The number of methods developed for these purposes has increased steadily in recent years and discussing them all is beyond the scope of this review. However, some of the limitations and strengths of these methods have been extensively reviewed elsewhere [84,120,121,122].

In recent years, a number of strategies based on the use of deep learning and complex network analyses have emerged to uncover the relevant dysregulations of lncRNAs and their potential value in identifying their relationships with cancer diagnosis, prognosis, and treatment [123,124,125,126,127]. One of the limitations of such algorithms is the lack of reliable and verified negative samples. In addition, they use known biomolecule–disease associations that cannot be applied to new diseases. Finally, these data integration tools, while useful for efficiently learning the structure of high-dimensional omics data, generally decrease the interpretability of the model, which is a limitation in a biomedical application where we wish to understand the relationship between the learned latent variables and the observed variables, such as disease. In recent years, formal work has begun to overcome this obstacle [128,129].

In addition, all the data have different calculation rules, which can lead to inaccuracies. Moreover, the link between lncRNAs and cancer goes beyond identifying the molecules that may be actively involved in the disease. Other aspects have been explored, such as the identification of biomarkers that are capable of predicting prognosis or even a response to drugs [130,131]. For this area, the combination of different approaches has been proposed, such as the least absolute shrinkage and selector operation (LASSO) and the support vector machine recursive feature elimination (SVM-RFE) [132,133].

In general, the methods for lncRNA identification, the prediction of their functions, and their relationship to disease, even as part of precision medicine tools for prognosis and drug response prediction, continue to improve by using higher level features and generating more and new information to feed the algorithms for better classifications. Despite these efforts, the integration of all the available information in a biologically meaningful way remains a challenge. During biological analysis, and more specifically in omics data analysis, subsequent functional validation steps are essential to provide new, more complete, and useful insights for future applications that can be transferred to the clinic. Thus, designs should include a holistic view integrating computational and experimental characterization.

## 8. Perspective on the New Directions in lncRNA Research and Its Implication in Cancer through Multi-Omics Analysis

What is the best way to answer new questions and discover the novel and controversial mechanisms of lncRNAs in a disease state through multidimensional analysis? A good example is the milestone discovery that lncRNAs are completely unable to produce peptides, even though there is evidence that some lncRNAs are found in the cytoplasm and can associate with ribosomes, suggesting the possibility of translation and the production of microproteins [134].

Unlike other protein by-products, such as insulin, which are derived from larger proteins, microproteins are small and translated directly from short open reading frames (sORFs) of less than 300 nucleotides that produce a protein of up to 100 amino acids. These sORFs are often found within the lncRNAs sequence. Despite the difficulties, researchers are making progress in characterizing the functions of the microproteins generated by lncRNAs that are altered in cancer and are related to the clinical behavior of tumors [135,136,137].

Although the complete repertoire of functional sORFs is unknown, new techniques and computational approaches are being developed based on the search for conserved DNA sequences, natural selection, ancient phylogenetic origins, and ribosomal profiling—a tool based on deep sequencing that allows for detailed measurements of translation [138,139,140]. Combining these methods, the lncRNA sORFs capable of encoding a microprotein have been identified. The data have shown that these microproteins are signaling molecules that regulate enzymes and act as receptor ligands and critical transmembrane components. More microproteins are being described which have implications in cancer. Here, we list some of the microproteins produced by lncRNAs linked to cancer (Table 2).

## 9. Perspective on the Future Use of lncRNAs for Therapeutic Purposes through Multi-Omics Oncology

The field of lncRNA research has developed rapidly over the past decade, moving from understanding their basic biological properties to exploiting their clinical relevance. As a result, lncRNAs have been proposed as biomarkers and therapeutic targets that are being actively explored. In this section, we review the emerging strategies for the therapeutic exploitation of lncRNA.

It would be appropriate to begin by discussing the value of circulating lncRNAs as biomedical tools for the detection and monitoring of various diseases. Despite the important progress in translating the clinical utility of circulating molecules as biomarkers, such as proteins, metabolites, or free mRNA, there are still relevant limitations. This is why the study of lncRNA has been postulated as a new source of biological information that presents relevant advantages over other circulating analytes, such as a greater resistance to degradation, stability in biofluids due to their secondary structures, and their transport in extracellular vesicles. These characteristics present them as reliable cancer biomarkers [148,149].

The case of PCA3 is a good example to illustrate the potential use of lncRNAs as informative biomarkers in cancer clinics. In 1999, Bussemakers and colleagues discovered that the ncRNA prostate cancer antigen 3 (PCA3), which they initially named differential display code 3 (DD3), was overexpressed in prostate cancer tissues compared to non-neoplastic prostate tissues [106]. Interestingly, PCA3 non-coding RNA is involved in the control of prostate cancer cell survival and modulates androgen receptor signaling [99]. Although PCA3 is the only lncRNA that has received FDA approval for the diagnosis, prognosis, or prediction of treatment response in any type of cancer, there are other reports describing new potential lncRNA biomarkers with high sensitivity and specificity for detecting specific neoplasms [150]. Thus, the diagnostic and monitoring utility of circulating lncRNAs biomarkers has not yet reached its full potential. For example, MIR205HG, the host gene of miR-205, is involved in prostate cancer cell differentiation and has been described as a potential marker that is capable of differentiating prostate cancer samples from normal ones [151].

A rapidly growing area of interest in the field of oncology is the phenomenon of drug resistance, which is the major limiting factor in achieving a cure for cancer patients. Emerging preclinical evidence supports that the lncRNA expression patterns predict a response to anticancer drugs [130,152]. By comparing the publicly available transcriptional profiles of different RNA species at baseline and after drug treatment with hundreds of compounds in the cancer cell lines, many lncRNAs such as GAS5 and ZEB2AS1 were shown to be highly predictive of the sensitivity to various anticancer drugs. Therefore, lncRNAs could explain new signals of how cancer cells become resistant to anticancer therapies and represent a new source of biomarkers. In addition, recent evidence showed that the lncRNAs EGFR-AS1 and MIR205HG could significantly improve the response prediction to erlotinib and gefinitib, better than EGFR somatic mutations and amplifications, suggesting a critical role for these lncRNAs in cancer precision medicine [153].

Given that lncRNAs can act as competitive endogenous RNAs (e.g., siRNAs or miRNA sponges), they could potentially control resistance-related biological processes. By constructing a ceRNA network, including lncRNA and mRNA, several drug resistance-related modules were identified as novel drug resistance markers. For example, the GAS5-RPL8 ceRNA pair regulated the drug resistance, as the GAS5 down-expression enhanced the RPL8 miRNA inhibition, which was reported to be associated with chemotherapy resistance [154]. The integration of multi-omics data revealed another relevant ceRNA module, consisting of HOXA-AS2, which was regulated by EIF4A3, FMR1, and HNRNPA2B1 and was down-expressed mainly in breast cancer patients, leading to adriamycin resistance. These alterations also down-expressed the expression of miR-107 [155]. Taken together, these studies demonstrated the potential viability of lncRNAs as complementary biomarkers and drug targets.

The therapeutic targeting of non-coding RNAs is an attractive approach for the treatment of cancer. Although no lncRNA-based therapy has been introduced into clinics, the functional diversity of lncRNAs provides an opportunity for their therapeutic modulation through transcriptional and post-transcriptional inhibition, the steric hindrance of promoters or secondary structures, exogenous synthetic lncRNAs, and editing tools such as CRISPR-Cas systems [122,123,156,157]. Each of these approaches has its own challenges, and future studies are needed to demonstrate their therapeutic efficacies.

## 10. Conclusions and Future Directions

With the advent of new technological resources, such as single-cell and spatial genomics and transcriptomics and the popularization of and ease of access to machine learning frameworks, multi-omics data integration approaches will further establish themselves as a powerful paradigm in the study of lncRNAs in cancer. The current state of knowledge includes a variety of studies and reviews that offer different perspectives on the topic. These range from studies that focus on elucidating the biological role of lncRNAs in cancer, to their proposed utility as molecular markers in clinical cancer research and the revision of computational methods for predicting lncRNA interactions with other molecules. However, many questions remain regarding the pragmatic application of the different molecular technologies and bioinformatics algorithms for mining multi-omics data. Furthermore, there is no clear consensus on how to integrate the different omics platforms and what information can be obtained from such approaches that can be used for the biological and pathological characterization of lncRNAs. Although many papers in the literature claim to have discovered the biological role or association between different lncRNAs and cancer using multi-omics strategies, only a fraction of them integrated information from different molecular platforms. Others explored different aspects of the same molecular features interrogated by each platform (e.g., RNA expression (gene expression, microRNAs expression), DNA somatic mutations, copy-number variations (CNV), or DNA methylation). In this review, we referred to multi-omics as the paradigm of using data obtained from and including two or more of the aforementioned molecular features. There is little doubt that these methods will complement our set of resources to unravel the relationships between lncRNAs and other biological factors that critically modulate cancer cell fitness and demonstrate new resources for studying tumor-relevant lncRNAs using dedicated multi-omics computational tools. It is highly desirable that, in the foreseeable future, researchers make efforts to become native users of these technologies to keep up with the current and future challenges in the field of cancer lncRNAs.

A crucial step in the study of lncRNAs as disease markers from a statistical point of view is the validation of their performance as predictors. However, there is still vague information and discrepancies about the biostatistical and machine learning approaches that have been used to discover and evaluate the performance of lncRNAs signatures, the power of these approaches, and the recall and sensitivity they report. In this regard, it is in the best interest of the entire research community to reach a consensus on the best practices for producing a more robust statistical model and description of the many steps in the selection of these models and validation methods. For example, the studies that used feature selection strategies to select lncRNAs as the variables of interest followed by the use of univariable or multi-variable regression models (e.g., Cox proportional hazards regression analysis) should provide additional rationales for the chosen method. In order to increase the scientific validity of statistical or machine learning approaches based on multi-omics data, it is important that researchers ensure the reproducibility of their findings by incorporating cross-validation information, which would give researchers a clearer perspective on the bias and general application of such models.

As the pace of technological advancement accelerates, an even greater amount of data will be generated from which researchers can extract novel features to describe the biological, physiological, and pharmacological properties of lncRNAs. Perhaps the two most notorious state-of-the-art paradigms in transcriptomic analysis are single-cell and spatial RNA-seq technologies, which seek to explore the repertoire of RNA molecules and their differential expressions at a single cell level and histological coordinates. Beyond the obvious cost of computational resources, this scenario comes at the expense of simple modeling, as incorporating these data into our research will add new layers of complexity. In this context, the lack of parsimony could become a more latent risk when generating new models explaining the associations between lncRNAs and different cancer hallmarks and their observable pathophysiological consequences (i.e., tumor invasion, metastasis, immune evasion, immune infiltration, etc.). By integrating single-cell and spatial transcriptomics data into multi-omics frameworks, researchers would be able to obtain new features such as the proportion and identity of cells within a given tumor that express a particular lncRNA. Even more, these data could be used to refine the association of lncRNA molecules with features such as the TMB, which is currently of value in cancer prognosis, the proportion of specialized cells harboring a somatic aberration in a locus containing functional lncRNAs, or the specific wiring of epigenomics interactions and signals in which lncRNAs are involved within such a cell. From a clinical point of view, having these features as the predictor variables associated with a particular outcome could indeed make the modeling process more demanding and complex, with the risk of overfitting. However, this risk can be mitigated if the precautions mentioned in the previous paragraphs are taken into account. Ultimately, the addition of these new dimensions could enhance the potential use of lncRNAs as diagnostic, prognostic, and predictive factors with more objective real-world applications in cancer clinics.

## Figures and Tables

**Figure 1 ijms-24-16600-f001:**
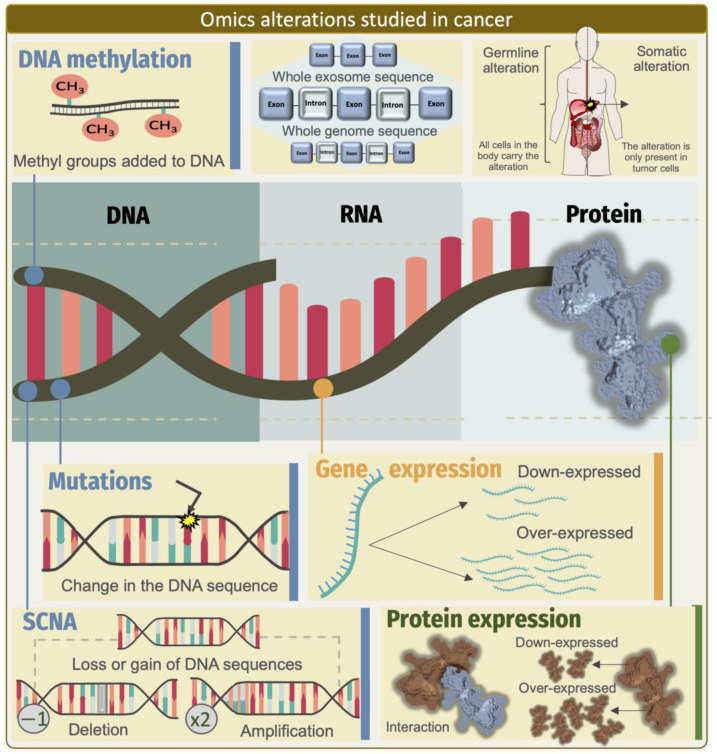
Representation of the various molecular features with changes and alterations at the DNA, RNA, and protein levels that can occur in cancer cells, representing the spectrum of multi-omics states that can be interrogated by high-throughput methods. DNA is the substrate for somatic aberrations, such as point mutations (arrow pointing at yellow spark) and somatic copy-number alterations (SCNA). DNA is also the molecule where the methylation of 5-methy-cytosine in different patterns has consequences in epigenomic regulation. On the other hand, the transcriptome, which is composed of the many RNA species in a cell, including the coding and non-coding genes (i.e., mRNAs, lncRNAs, and microRNAs), undergoes various dynamic changes that affect the levels of these species in different cells and tissues. Changes in the expression of various RNA molecules, such as lncRNAs, have become a potential repertoire of functional molecular markers for different types of cancer. Similarly, proteins and metabolites can also become informative indicators of the oncogenic processes occurring in a cell, tissue, organ, or physiological system in the body. Although graphically represented as separate compartments, these molecular dimensions are interconnected, and changes in the state in one dimension have consequences in the others and vice versa.

**Figure 2 ijms-24-16600-f002:**
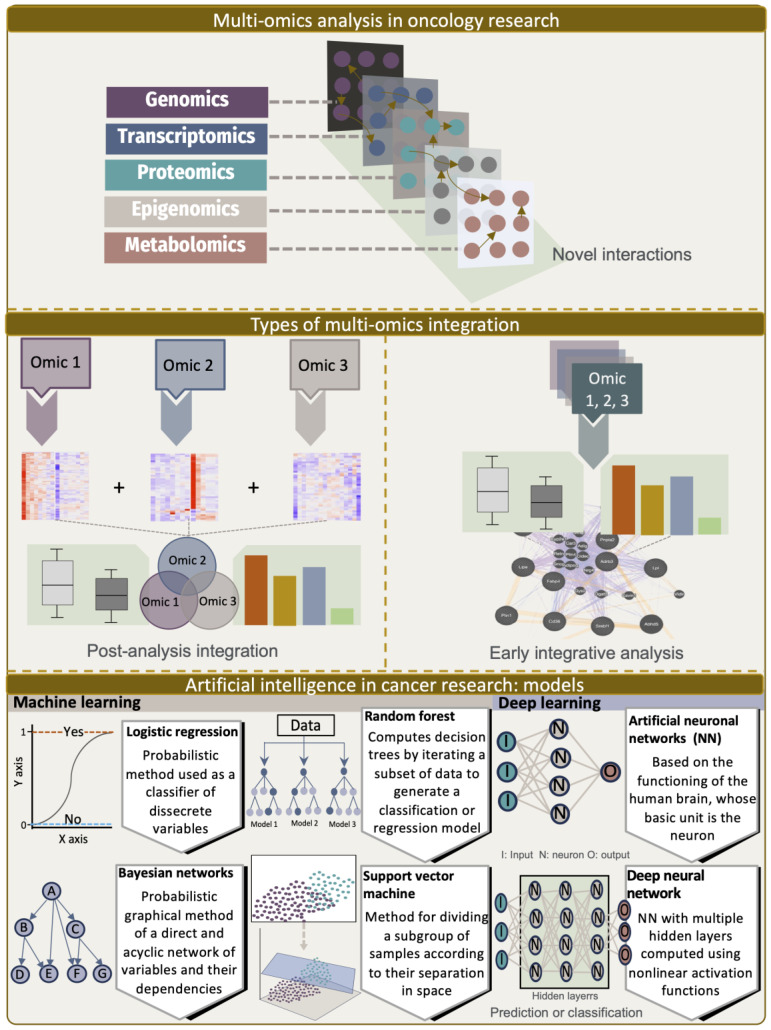
Multi-omics approaches used in cancer research and interrogated at a multidimensional level for the characterization of the biological, functional, and pathophysiological properties of lncRNA. In multi-omics, different modalities for the study design have been proposed, mainly combining two or more dimensions of molecular features (i.e., genomics, transcriptomics, epigenomics, proteomics, and metabolomics). For that purpose, bioinformatics and statistical frameworks have been also envisaged by researchers to mine information from the integrative analysis of these multidimensional data structures (e.g., post-analysis and early integration analysis). In combination with bioinformatics pipelines, different statistical and machine learning frameworks have been implemented, which are aimed at constructing and refining models to predict the outcome of cancer patients, such as free survival, time to relapse, and treatment response.

**Figure 3 ijms-24-16600-f003:**
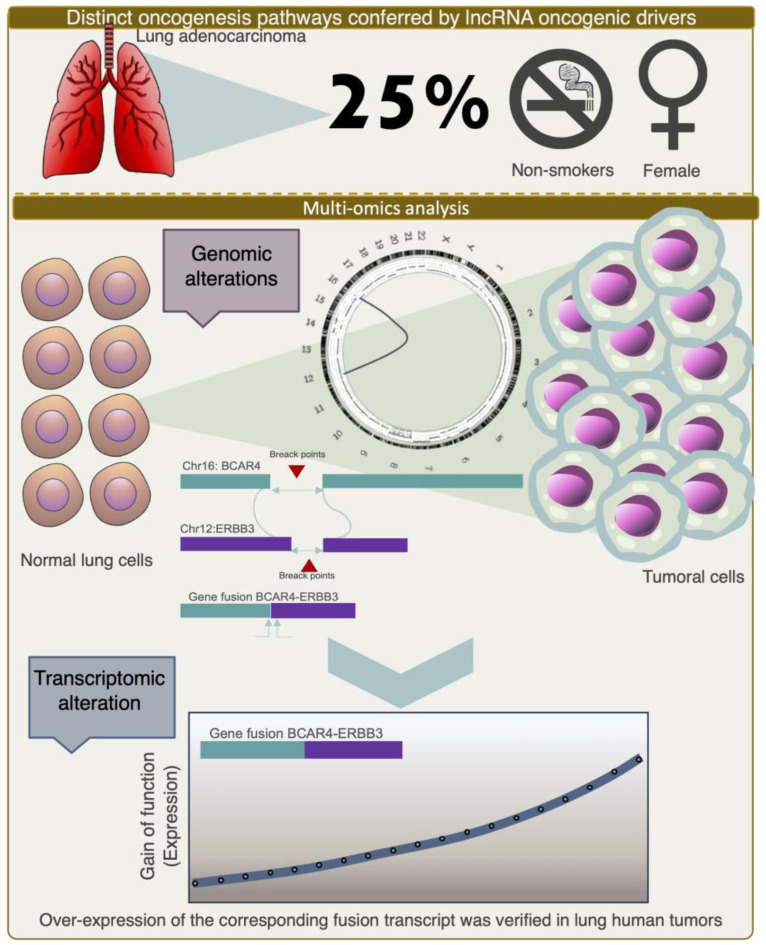
Summary of an integrative multi-omics analysis in lung adenocarcinoma to address how the complex genomic rearrangements and the resulting gene fusions, which are often copy-number balanced, appear as a process of early oncogenesis that is often acquired in the first decades of life. We graphically depicted an alteration described in 25% of lung adenocarcinoma cases without a history of smoking, mainly enriched in female patients and whose etiology is unknown (**upper** panel). The integrative multi-omics analysis identified a recurrent oncogenic fusion gene composed of BCAR4:ERBB3 (lncRNA and messenger RNA) that was also overexpressed in a subset of cases (**bottom** panel).

**Figure 4 ijms-24-16600-f004:**
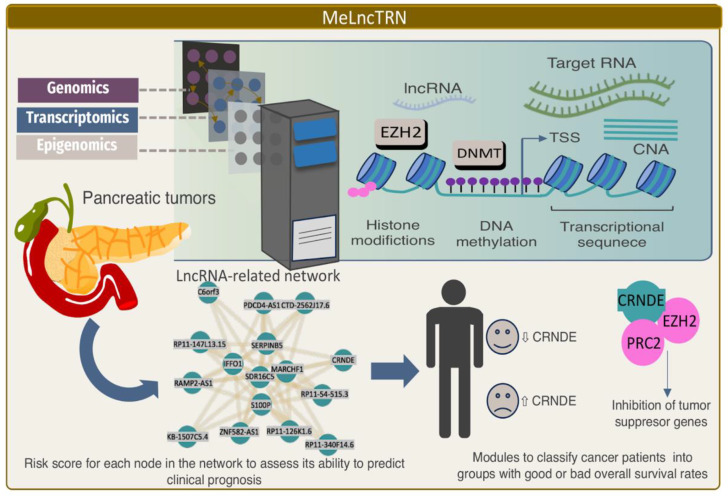
Summary representation of the multi-dynamic analysis computed by MeLncTRNn, which integrated genomic, transcriptomic, and epigenetic data to identify lncRNA genes and their interaction with epigenetic mechanisms that could serve as markers for classifying cancer patients into subtypes with different prognoses (**top** panel). We described a regulatory network operating in pancreatic cancer defined by MeLncTRNn, integrating lncRNA modulators and target genes related to epigenetic regulation. In particular, the CRNDE module serves as a prognostic information tool for pancreatic cancer patients (**bottom** panel).

**Table 1 ijms-24-16600-t001:** Research studies conducted, in which multi-omics approaches were used to identify and characterize the biological and clinico-pathological features of different lncRNAs signatures and evaluate their performance as predictors of patient clinical outcomes for various cancer types. NA: No information available.

lncRNA of Relevance from the Study	Potential Association Based on Biological or Clinical Data	Goal	Main Insight Regarding the lncRNA	Model Construction	Kaplan–Meier and Log-Rank Test	Receiver Operator Curve AUC	Reference
ESR1, TRPS1, ERG, RUNX1, SNHG16, and HOTAIR	Various cancer types	Molecular and biological characterization of the regulatory mutations on functional lncRNAs in cancer	Somatic mutations on the lncRNA TF binding site are associated with their expression and activity in cancer	NA	NA	NA	[32]
CASC8	Breast cancer	Prioritize and predict lncRNAs with a functional impact through mutation analysis	Somatic mutations on the lncRNA loci site are associated with expression alterations that have a functional impact	NA	NA	NA	[37]
ENSG0000021403, ENSG00000261650, ENSG00000281406, and G001643	Colorectal carcinoma	Molecular and biological characterization of the regulatory mutations on functional lncRNAs in cancer	Somatic mutations on the lncRNA loci site are associated with expression alterations that have a functional impact	NA	NA	NA	[38]
LINC00460, AC156455.1, AC015977.2, ‘PRDM16-dt’, AL139351.1, AL035661.1, and LINC01606	Renal cell carcinoma	Identify the genome instability-related lncRNAs and their clinical significance	Discovery of 11 lncRNAs related to mutational burden and associated with patient poor overall survival	Univariable and multi-variable COX prognostic regression model	High-risk vs. low-risk groups; *p* < 0.001	Train AUC = 0.743 Test = 0.770	[39]
LINC00460, LINC01234	Clear cell renal carcinoma	Identify the genome instability-related lncRNAs and their clinical significance	Identification of a lncRNAs signature related to the mutational burden and associated with patient poor overall survival	Univariable and multi-variable COX prognostic regression model	High-risk vs. low-risk groups; *p* < 0.001	AUC = 0.681	[40]
FAM30A, CACNA1C-AS1, LINC02595, LINC00926, AL589863.1, and AP000919.3	AML	Construct a somatic mutation-associated risk index	lncRNAs related to the mutational burden and associated with patient poor overall survival	Selection of candidate lncRNAs using LASSO followed by the univariable and multi-variable COX prognostic regression model	High-risk vs. low-risk groups; *p* < 0.001	AUC = 0.804	[41]
AC007996.1, AC009237.14, AP003555.1, and AL590483.1	Colorectal carcinoma	Evaluate the performance of a genome stability-related lncRNA signature as a risk predictor	Identification of a lncRNAs signature related to the mutational burden and associated with patient poor overall survival	Univariable and multi-variable COX prognostic regression model	High-risk vs. low-risk groups at 3 years; *p* < 0.001	AUC = 0.713	[42]
ZNF503-AS1, AL353747.2, AC129492.1, AP003555.1, and AC009237.14	Colorectal carcinoma	Evaluate the performance of a genome stability-related lncRNA signature as a risk predictor	Identification of a lncRNAs signature related to the mutational burden and immune infiltration that are associated with patient poor overall survival	Univariable and multi-variable COX prognostic regression model	High-risk vs. low-risk groups; *p* < 0.001; in the validation set	AUC (1 year) = 0.750; AUC (3 years) = 0.757, AUC (5 years) = 0.711; in the validation set	[43]
AC107464.2, MIR100HG, and AP001527.2	Cervical carcinoma	Evaluate the performance of a genome stability-related lncRNA signature as a risk predictor	Identification of a lncRNAs signature related to the mutational burden and associated with patient poor overall survival	Univariable and multi-variable COX prognostic regression model	High-risk (1.7 years) vs. low-risk (1.5 years) groups; *p* < 0.001; in the validation set	AUC (3 years) = 0.663; in the validation set	[44]
AC002511.2, LINC00501, LINC02055, LINC02714, LINC01508, LOC105371967, RP11_96A15.1, RP11_305F18.1, RP11_342M1.3, RP11_432J24.3, and U95743.1	Hepatocellular carcinoma	Evaluate the performance of a genome stability-related lncRNA signature as a risk predictor	Identification of a lncRNAs signature related to the mutational burden and associated with patient poor overall survival	Selection of candidate lncRNAs using LASSO followed by the univariable and multi-variable COX prognostic regression model	High-risk vs. low-risk groups; *p* < 0.001	NA	[45]
C116351.1, ZFPM2-AS1, AC145343.1, and MIR210HG	Hepatocellular carcinoma	Evaluate the performance of a genome stability-related lncRNA signature as a risk predictor	Identification of a lncRNAs signature related to the mutational burden and associated with patient poor overall survival	Univariable and multi-variable COX prognostic regression model	High-risk vs. low-risk groups at 3 and 5 years; *p* < 0.001	AUC (3 years) = 0.710, AUC (5 years) = 0.707; in the validation set	[46]
BCAL8	Breast cancer	Molecular and biological characterization of the functional regulatory mutations on cancer	Identification of a somatic SCNA-related lncRNAs signature associated with patient prognosis	Selection of candidate lncRNAs using filtering methods	NA	NA	[47]
RUSC1-AS1, LINC01990, LINC01411, LINC02099, H19, LINC00452, ADPGK-AS1, and C1QTNF1-AS1	Cervical carcinoma	Evaluate the performance of a SCNA-related lncRNA signature as a risk predictor	Identification of a somatic SCNA-related lncRNAs signature with a value as independent tumor-free survival	Univariable and multi-variable COX prognostic regression model	High-risk vs. low-risk groups at 1, 3, and 5 years; *p* < 0.001	AUC (1 year), AUC (3 years), and AUC (5 years) > 0.750	[48]
PRAL	Tumor suppressor	Molecular and biological characterization of SCNA-related lncRNAs in cancer	Identification of a somatic SCNA-related lncRNA with a value as an independent predictor for reduced tumor-free survival	Univariable and multi-variable COX prognostic regression model	High-risk vs. low-risk groups; *p* < 0.001	NA	[49]
LOC101927604, LOC105377267, CASC15, LINC-PINT, CLDN10-AS1, C14orf132, LMF1, LINC00675, CCDC144NL-AS1, and LOC284454	Colorectal carcinoma	Evaluate the performance of an SCNA-related lncRNA signature as a risk predictor for CRC	Identification of somatic SCNA-related lncRNA with a value as an independent predictor for reduced tumor-free survival	Clustering based on the gene expression, SCNA, and DNA methylation followed by the univariable and multi-variable COX prognostic regression model	High-risk vs. low-risk groups; *p* < 0.001	NA	[50]
RP11-571M6.8	Glioblastoma	Molecular and biological characterization of SCNA-related lncRNAs in cancer	Identification of a somatic SCNA-related lncRNA that is significantly predictive of disease-free survival	Selection of candidate lncRNAs using filtering methods followed by the univariable and multi-variable COX prognostic regression model	High-risk vs. low-risk groups; *p* < 0.001	NA	[51]
RP11-1020A11.1	Bladder carcinoma	Molecular and biological characterization of SCNA-related lncRNAs with risk prediction utility in cancer	Identification of a somatic SCNA-related lncRNA that is significantly predictive of disease-free survival	Selection of candidate lncRNAs using filtering methods followed by the univariable and multi-variable COX prognostic regression model	High-risk vs. low-risk groups; *p* < 0.001	NA	[51]
LINC02528, SEMA6A-AS1, EBLN3P, MIR155HG, LYRM4-AS1, and HLA-DQB1-AS1	Skin cutaneous melanoma	Identify a novel prognostic signature using m6A-related lncRNAs and evaluate the prognostic of survival performance	Identification of a lncRNAs signature-related somatic SCNA associated with patient overall survival	Selection of candidate lncRNAs using LASSO followed by the COX prognostic regression model	High-risk vs. low-risk groups at 1, 2, 3, and 5 years; *p* < 0.001 and construction of a nomogram for the clinical decision risk score	AUC (1, 2, 3, and 5 years) > 0.6	[52]
AL121772.1, BX640514.2, LINC01133, and LYPLAL1-AS1	Pancreatic cancer	Investigate the prognostic performance of a lncRNA signature and its relationship with the tumor immune microenvironment	Identification of a genomic instability-related lncRNAs signature associated with patient overall survival	Selection of candidate lncRNAs using filtering methods followed by the univariable and multi-variable COX prognostic regression model	High-risk vs. low-risk groups at 1 year; *p* < 0.001	AUC (1 year) = 0.653	[53]
CTD-2256P15.2	Lung adenocarcinoma	Molecular and biological characterization of SCNA-related lncRNAs with risk prediction utility in cancer	Identification of a somatic SCNA-related lncRNAs signature associated with the prognosis of patients after methyl ethyl ketone (MEK) inhibitors treatment	Selection of candidate lncRNAs using filtering methods followed by the univariable and multi-variable COX prognostic regression model	High-risk vs. low-risk groups; *p* < 0.001	NA	[54]
LINC00896, MCM8-AS1, LINC01251, LNX1-AS1, GPRC5D-AS1, CTD-2350J17.1, LINC01133, LINC01121, and AC073130.1	Non-small cell lung cancer	Evaluate the performance of a SCNA-related lncRNA signature as a risk predictor	Identification of a lncRNAs signature related to somatic SCNA and associated with patient poor overall survival	Selection of candidate lncRNAs using LASSO followed by the COX prognostic regression model	High-risk vs. low-risk groups at 1 and 3 years; *p* < 0.001	AUC (1 and 3 years) = 0.73	[54]
RP11-241F15.10	Osteosarcoma	Evaluate the performance of a SCNA-related lncRNA signature as a risk predictor	Identification of a lncRNAs signature-related somatic SCNA associated with patient disease-free and overall survival	Univariable and multi-variable COX prognostic regression model	High-risk vs. low-risk groups at 1 and 3 years; *p* < 0.001	NA	[55]
ALAL-1	Non–small cell lung cancer	Molecular and biological characterization of SCNA-related lncRNAs in cancer	Identification of a pro-oncogenic lncRNA that mediates cancer immune evasion, pointing to a new target for immune potentiation	NA	NA	NA	[56]
LOC339803, F11-AS1, and PCAT2 TMEM220-AS1	Hepatocellular carcinoma	Evaluate the performance of a SCNA-related lncRNA signature as a risk predictor	Identification of CNA-related lncRNAs that can better evaluate the prognosis of patients with liver cancer	Selection of candidate lncRNAs using LASSO followed by the COX prognostic regression model	High-risk vs. low-risk groups at 1, 3, and 5 years; *p* < 0.001	AUC (1, 3, and 5 year) > 0.7	[57]
ENSG00000261582	Lung adenocarcinoma and cervical carcinoma	Evaluate the performance of a SCNA-related lncRNA signature as a risk predictor	Identification of a somatic SCNA-related lncRNAs signature associated with patient overall survival	Univariable and multi-variable COX prognostic regression model	High-risk vs. low-risk groups at 5 years; *p* < 0.001	NA	[58]
PCAN-R1 (Ensembl ID ENSG00000228288) and PCAN-R2 (Ensembl ID ENSG00000231806)	Prostate adenocarcinoma	Evaluate the performance of a SCNA-related lncRNA signature as a risk predictor	Identification of a somatic SCNA-related lncRNAs signature associated with patient overall survival	Univariable and multi-variable COX prognostic regression model	High-risk vs. low-risk groups at 5 years; *p* < 0.001	NA	[58]
LOC101927151, LINC00861, and LEMD1-AS1	Ovarian cancer	Evaluate the performance of a SCNA-related lncRNA signature as a risk predictor	Identification of a somatic SCNA-related lncRNAs signature associated with patient prognosis	Univariable and multi-variable COX prognostic regression model	High-risk vs. low-risk groups at 5 years; *p* < 0.001	NA	[59]
TSPOAP1-AS1, CCNT2-AS1, LINC01094, AL033527.2, and LINC00460	Gastric cancer	Evaluate the performance of a mutation-related lncRNA signature as a risk predictor and characterization of functional activity	Identification of an anoikis-related lncRNAs signature associated with patient poor overall survival and immunotherapy response	Selection of candidate lncRNAs using LASSO followed by the COX prognostic regression model	High-risk vs. low-risk groups; *p* < 0.001	Train AUC = 0.707 Test = 0.646	[53]
NA	Kidney renal clear cell carcinoma	Evaluate the performance of a multi-omics-derived lncRNA signature as a risk predictor	Identification of a lncRNAs signature derived from the analysis of transcriptomics and DNA methylation associated with patient poor overall survival	Selection of candidate lncRNAs using a novel TRS method utilizing multiple omics data and a XGBoost model followed by the COX prognostic regression model	High-risk vs. low-risk groups; *p* < 0.001	AUC = 0.95; using the best predictor model	[60]
Twenty-six lncRNAs	Hepatocellular carcinoma	Evaluate the performance of a multi-omics-derived lncRNA signature as a risk predictor and characterization of functional activity	Identification of an exosome-related lncRNAs signature associated with patient poor overall survival and a response to transarterial chemoembolization (TACE) therapy and sorafenib therapy	Selection of candidate lncRNAs using a weighted correlation network analysis followed by the COX prognostic regression model	High-risk vs. low-risk groups; *p* < 0.001	AUC > 0.7	[61]
Ten lncRNAs, including: LINC00582, MIR205HG and TRG-S1,	Breast cancer	Evaluate the performance of a mutation-related lncRNA signature as a risk predictor and characterization of functional activity	Identification of a TMB-related lncRNAs signature associated with patient poor overall survival and a response to immunotherapy	Clustering based on the gene expression and selection of the gene predictors using LASSO followed by the COX prognostic regression model	High-risk vs. low-risk groups at 1, 3, and 5 years; *p* < 0.001	AUC (1 year) = 0.722, AUC (3 years) = 0.745, AUC (5 years) = 0.811	[62]
C20orf197, UCA1, MIR17HG, and MIR22HG	Various cancer types	Evaluate the performance of a lncRNA signature as a patient prognosis predictor	Oxiplatin sensitivity-related lncRNAs signature associated with the prognosis of patients given oxaliplatin-based chemotherapy	Selection of candidate lncRNAs using LASSO, decision tree, random forest, and a support vector machine followed by the COX prognostic regression model	High-risk vs. low-risk groups at 1, 3, and 5 years; *p* < 0.001	AUC (1 year) = 0.76, AUC (3 years) = 0.79, AUC (5 years) = 0.88; using the best predictor model	[63]
18 lncRNAs	Gastric cancer	Evaluate the performance of a lncRNA signature as a patient prognosis predictor	Identification of an immune-related lncRNAs signature that helps predict the prognosis of patients suffering from gastric cancer	Selection of candidate lncRNAs using the integration of multiple machine learning algorithms followed by the COX prognostic regression model	High-risk vs. low-risk groups at 1, 3, and 5 years; *p* < 0.001	AUC (1 year) = 0.715, AUC (3 years) = 0.80, AUC (5 years) = 0.809; using the best predictor model	[64]
Various lncRNAs	Breast cancer	Develop an interpretable deep-learning-based network for classifying the recurrence risk and revealing the potential biological mechanisms	Construction of a lncRNa-based model associated with the radiomics of magnetic resonance features for predicting individual recurrence risk after surgery	Construction of a lncRNa-based model using the Cox proportional hazards deep neural network	High-risk vs. low-risk groups at 1, 2, and 3 years; *p* < 0.001	AUC (1 year) = 0.98, AUC (2 years) = 0.94, AUC (3 years) = 0.92; using the best predictor model	[65]
CRNDE, AC010273.2, MPPED2-AS1, SNHG18, and CYTOR	Lower-grade gliomas	Evaluate the impact of DNA methylation-related lncRNAs with an effect on genome stability and the immune microenvironment on disease progression	Identification of a five DNA methylation-related signature with an independent prognostic value	Selection of lncRNAs using filtering methods followed by the COX prognostic regression model	High-risk vs. low-risk groups at 1, 2, and 3 years; *p* < 0.001	AUC (1 year) = 0.893, AUC (2 years) = 0.919, AUC (3 years) = 0.866; using the best predictor model	[66]
CRNDE, AC010273.2, MPPED2-AS1, SNHG18, and CYTOR	Various cancer types	Explore the functional effects of lncRNAs related to DNA methylation and evaluate their predictive performance on patient survival	Identification of a DNA methylation and genome stability-related signature with an independent prognostic value	Construction of a lncRNa-based co-expression network of pADM–lncRNA	High-risk vs. low-risk groups; *p* < 0.001	AUC = 0.6839	[67]
LIFR-AS1	Colorectal carcinoma	Evaluate the impact of DNA methylation-related lncRNAs with an effect on disease progression	Identification of LIFR-AS1 as a tumor suppressor RNA with an independent prognostic value	Selection using filtering methods followed by the COX prognostic regression model	High-risk vs. low-risk groups; *p* < 0.001	AUC = 0.872	[68]

**Table 2 ijms-24-16600-t002:** Studies focused on describing the microproteins translated from the lncRNA sequences with relevant implications in cancer.

lncRNA	Micropeptide	Biological Effect	Method	Citation
lncRNA AC025154.2	MIAC	Inhibits the actin cytoskeleton by interacting with Aquaporin 2 to suppress tumor growth and metastasis	RNA-seq data from kidney renal clear cell carcinoma patients	[141]
LINC-PINT	PINT87aa	Suppresses glioblastoma cell proliferation by interacting with PAF1c and preventing the transcriptional elongation of cancer-related genes	RNA-seq of circRNAs (transcriptome sequencing) and RNC-RNAs (translatome sequencing)	[142]
LINC02381	LINC02381-aa	LINC02381-aa contained in exosomes enhances ferroptosis by promoting the glucose transporter SLC2A10 in glioblastoma	Machine learning to integrate the multi-omics data to identify microproteins	[143]
NR_029453	CASIMO1	Interacts with the enzyme squalene epoxidase and increases the phosphorylation of ERK, a relevant actor of the MAPK pathway affecting cell proliferation	Transcriptome analysis and DNA conservation evaluation	[144]
lncRNA CTD-2256P15.2	PACMP	PACMP acts as an activator of the PARP1-dependent pathways (DNA repair process), enhancing tumor growth and limiting the cell response to PARP inhibitors	RNA-seq	[145]
HOXB-AS3	HOXB-AS3 peptide	Suppresses cancer growth by limiting PKM splicing and, subsequently, the metabolic reprogramming	Transcriptome analysis and RNA affinity purification analysis	[146]
LINC00665	CIP2A-BP	Decreases cell invasion in vivo and correlates with better patient survival. Competes with PP2A for CIP2A binding, suppressing the oncogenic P13K/AKT/NFkB pathway	Ribosome profiling and RNA sequencing data analysis	[147]

## Data Availability

Not applicable.

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
