# Peer review of "Multi-Omics Mining of lncRNAs with Biological and Clinical Relevance in Cancer"

_ijms, 2023, doi:10.3390/ijms242316600_

Round 1

Reviewer 1 Report

Comments and Suggestions for Authors

This review offers a broad overview of long non-coding RNAs (lncRNAs) and their existing and potential role as biological markers in cancer. The manuscript investigates instances of multi-omics approaches that merge data on somatic aberrations, gene expression, and epigenomics. While the review is comprehensive for several cancer associations with lncRNAs. The validation of those biomarkers needs more highlighting.

I suggest investigating those findings through GO-enrichment, KEGG pathway analysis, or any pathway analysis database.

Author Response

Comments and Suggestions for Authors

This review offers a broad overview of long non-coding RNAs (lncRNAs) and their existing and potential role as biological markers in cancer. The manuscript investigates instances of multi-omics approaches that merge data on somatic aberrations, gene expression, and epigenomics. While the review is comprehensive for several cancer associations with lncRNAs. The validation of those biomarkers needs more highlighting.

I suggest investigating those findings through GO-enrichment, KEGG pathway analysis, or any pathway analysis database.

First of all, we would like to thank the reviewer for taking the time to read our manuscript and we appreciate his/her valuable recommendations. In accordance with them, we have made several changes and corrections, including the addition of new figures, one depicting the nature of the various molecular changes discussed and the other representative of the relevant biological function of some lncRNAs. We had previously considered performing an overrepresentation and gene enrichment analysis to describe the functional properties for different lncRNAs, but given their nature as non-coding molecules, efforts to generate catalogs of databases of biological or molecular terms remain somewhat scarce and very few resources are available to retrieve reproducible functional annotation of these molecules. Instead, we resort to a manual investigation of biological activity in the consulted literature and address the ones which in our opinion reflect some of the most relevant examples of biological activity of lncRNAs in cancer.

Reviewer 2 Report

Comments and Suggestions for Authors

The review article titled "MULTI-OMICS MINING OF LNC-RNAS WITH

BIOLOGICAL AND CLINICAL RELEVANCE IN CANCER.” It is interesting, but:

Structural Enhancements:

Tables: Considering the depth and complexity of multi-omics approaches and their potential applications, tables would be invaluable. These can describe the major lncRNAs studied, summarize their functional roles, and provide a comparative overview of methods, applications, and sensitivity limits.

Depth of Content:

Metabolite specificity: to provide readers with a holistic understanding, it would be beneficial to elaborate on the nature of the metabolites studied. In addition, explaining the exact methods used for their analysis would provide comprehensive insight.

New contributions and future directions:

Clarification of new insights: A key aspect of any review paper is to highlight the new perspective or synthesis that the article contributes to the existing literature. It's critical to articulate how this review differs from previous work. Is there a unique perspective or new synthesis of data being presented?

Future trends: a forward-looking section discussing potential trends, upcoming challenges, and areas to explore in the field of lncRNAs in cancer would add significant value. This not only helps researchers understand the evolution of the field, but also provides a roadmap for potential areas of investigation.

Visual Enhancements:

Graphical Summary: The inclusion of a graphical diagram or summary allows readers to quickly grasp the central themes and methods of the report. Visuals can efficiently convey complex concepts, methods, and potential applications of lncRNAs in cancer.

Clarity and flow:

Streamlined integration: when discussing multi-omics approaches, it may be helpful to more seamlessly integrate information on somatic aberrations, gene expression, and epigenomics. This may provide the reader with a smooth and coherent understanding of the landscape.

By considering these suggestions, the manuscript can further increase its utility and clarity for the target audience and ensure that the review not only provides an up-to-date overview but also serves as a guidepost for future research efforts in the field.

Author Response

We are grateful that the reviewer took the time to read our work and to provide comments and recommendations. We would like to provide a point-by-point response to his major concerns.

Comments and Suggestions for Authors

The review article titled "MULTI-OMICS MINING OF LNC-RNAS WITH

BIOLOGICAL AND CLINICAL RELEVANCE IN CANCER.” It is interesting, but:

Structural Enhancements:

Tables: Considering the depth and complexity of multi-omics approaches and their potential applications, tables would be invaluable. These can describe the major lncRNAs studied, summarize their functional roles, and provide a comparative overview of methods, applications, and sensitivity limits.

Thank you for your suggestion. We have now modified and reformatted the tables presented in our manuscript to better convey the focus of our work. In particular, we have included a comparative table summarizing the main methods used for each study and reported data on the use of lncRNA-based prediction models and the metrics used to evaluate their performance. In addition, we have made several structural changes to our manuscript to better reflect these findings.

Depth of Content:

Metabolite specificity: to provide readers with a holistic understanding, it would be beneficial to elaborate on the nature of the metabolites studied. In addition, explaining the exact methods used for their analysis would provide comprehensive insight.?

We have included a figure 1, 3 and 4 describing the different molecular characteristics (i.e. somatic mutations, CNV, DNA methylations, gene expression, etc) analyzed by the various “omics” technologies we cover in our review. 

New contributions and future directions:

Clarification of new insights: A key aspect of any review paper is to highlight the new perspective or synthesis that the article contributes to the existing literature. It's critical to articulate how this review differs from previous work. Is there a unique perspective or new synthesis of data being presented?

Again, thank you for your comment, we have now explained the relevance of the topic and the perspective we are trying to provide through the discussion along our manuscript. Especially in the Abstract, Introduction and Conclusion, and Future Perspectives sections.

Future trends: a forward-looking section discussing potential trends, upcoming challenges, and areas to explore in the field of lncRNAs in cancer would add significant value. This not only helps researchers understand the evolution of the field, but also provides a roadmap for potential areas of investigation.

In this regard, we have included a more comprehensive analysis of the utility and future vision of the topic. In particular, we have reorganized some sections in our manuscript, which we now present as: "Challenges and opportunities of machine learning to deepen the functional and biological roles of lncrna in cancer", "Perspective on new directions in lncrna research and its implication in cancer through multi-omics analysis", "Perspective on the future use of lncrna for therapeutic purposes through multi-omic oncology", and "Conclusions and future directions". In each of these sections, we attempt to provide a perspective on the progress and achievements in the field and the future trends.

Visual Enhancements:

Graphical Summary: The inclusion of a graphical diagram or summary allows readers to quickly grasp the central themes and methods of the report. Visuals can efficiently convey complex concepts, methods, and potential applications of lncRNAs in cancer.

Thank you for this suggestion. Accordingly, our review has been supplemented with 4 different figures to summarize important concepts and a graphical summary to convey the main perspective of our work.

Clarity and flow:

Streamlined integration: when discussing multi-omics approaches, it may be helpful to more seamlessly integrate information on somatic aberrations, gene expression, and epigenomics. This may provide the reader with a smooth and coherent understanding of the landscape.

As noted above, we have undertaken extensive reformatting and reorganization of various sections of our manuscript in order to provide a clearer and more integrative narrative of the main concepts and discussion, and to emphasize the main perspectives and insights we are trying to convey with our work.

By considering these suggestions, the manuscript can further increase its utility and clarity for the target audience and ensure that the review not only provides an up-to-date overview but also serves as a guidepost for future research efforts in the field.

We value your expertise and your valuable input and recommendations.

Reviewer 3 Report

Comments and Suggestions for Authors

Manuscript proposed by Salido-Guadarrama and co-workers entitled “MULTI-OMICS MINING OF LNC-RNAS WITH BIOLOGICAL AND CLINICAL RELEVANCE IN CANCER” is a review of the current panorama on lncRNAs with an actual or potential role as biological markers in cancer

In my opinion, the manuscript needs plenty of improvements/modifications because it is incomplete.

My major comments are presented below.

Major concerns:

- Abstract – please determine the importance of the topic and the novelty of this review

- What is the novelty of the presented work comparing previously presented reviews on this topic

- the reference list does not include all important literature position in this topic and last reviews

- Changes in the text are needed

- Check and correct English

Comments on the Quality of English Language

In my opinion, moderate English editing is needed. 

Author Response

Comments and Suggestions for Authors

Manuscript proposed by Salido-Guadarrama and co-workers entitled “MULTI-OMICS MINING OF LNC-RNAS WITH BIOLOGICAL AND CLINICAL RELEVANCE IN CANCER” is a review of the current panorama on lncRNAs with an actual or potential role as biological markers in cancer.

In my opinion, the manuscript needs plenty of improvements/modifications because it is incomplete.

My major comments are presented below. 

We are grateful that the reviewer took the time to read our work and to provide comments and recommendations. We would like to provide a point-by-point response to his major concerns. 

Major concerns:

- Abstract – please determine the importance of the topic and the novelty of this review

We have completely revised our manuscript to better reflect the relevance and novelty of our work. In particular, we have now added a paragraph stating what we believe to be the main and unique perspective of our work and its importance in addressing the topic we are presenting.

- What is the novelty of the presented work comparing previously presented reviews on this topic

Thank you again. As mentioned, we have enriched several sections of our manuscript to highlight the key findings of our discussion. In particular, we have provided more insight into the pertinence and relevance of the topics we cover, and why we believe we provide insight into "practical applications using different integrative, comparative, and multidimensional study designs incorporating bioinformatics and statistical analysis of multi-omics data" with a perspective not previously presented in the literature. We have supplemented our manuscript with a more comprehensive statement of the evolution of knowledge in this field in the Introduction, and in the final sections we discuss and present current and future challenges and solutions to be explored.

- the reference list does not include all important literature position in this topic and last reviews

We welcome this comment. The reference list has been thoroughly revised and we have updated it to include the more relevant literature. To clearly state our focus on the literature to be reviewed, we have now added a paragraph in the Introduction stating: "we searched for literature in the PubMed database using the keywords ("lncRNA" or "long non-coding RNA" and "cancer" and "multiomics") up to October 2023. Subsequently, we manually revised to focus mainly on different approaches for multi-omics integration with the scope of finding and evaluating functional lncRNAs signatures with clinical prognosis and prediction of patient outcome”.

- Changes in the text are needed

- Check and correct English

The entire manuscript has been thoroughly re-examined and corrected, both grammatically and orthographically.

We are entirely thankful for your expertise and for your valuable input and recommendations.

Round 2

Reviewer 2 Report

Comments and Suggestions for Authors

The Authors have performed the alterations.

Reviewer 3 Report

Comments and Suggestions for Authors

The revised version of the manuscript presented by Salido-Guadarrama and co-workers (ijms-2664217) entitled MULTI-OMICS MINING OF LNC-RNAS WITH BIOLOGICAL AND CLINICAL RELEVANCE IN CANCER meets my requirements. The authors have provided appropriate responses and comments to all of my comments. The text has been properly modified, errors have been corrected, and the text has been supplemented and formatted accordingly. The manuscript can be accepted for publication in the presented form. 

Comments on the Quality of English Language

I believe that the quality of the English is satisfactory. Some minor editing is required.